# Mental Health, Sleep, and Caffeine Intake Among Shift Workers in a Nationally Representative Sample of the Korean Adult Population

**DOI:** 10.3390/nu17071155

**Published:** 2025-03-26

**Authors:** Gyu-Lee Kim, Jinmi Kim, Jeong-Gyu Lee, Young-Jin Tak, Seung-Hun Lee, Young-Jin Ra, Sang-Yeoup Lee, Young-Hye Cho, Eun-Ju Park, Youngin Lee, Jung-In Choi, Yu-Hyeon Yi

**Affiliations:** 1Department of Family Medicine and Medical Research Institute, Pusan National University Hospital, Busan 49241, Republic of Korea; happygaru@hanmail.net (G.-L.K.); eltidine@hanmail.net (J.-G.L.); 03141998@hanmail.net (Y.-J.T.); greatseunghun@hanmail.net (S.-H.L.); yjra80@naver.com (Y.-J.R.); 2Department of Family Medicine, Pusan National University School of Medicine, Yangsan 50612, Republic of Korea; saylee@pnu.edu (S.-Y.L.); younghye82@naver.com (Y.-H.C.); everblue124@daum.net (E.-J.P.); ylee23@gmail.com (Y.L.); s1jungin@hanmail.net (J.-I.C.); 3Department of Biostatistics, Clinical Trial Center, Biomedical Research Institute, Pusan National University Hospital, Busan 49241, Republic of Korea; jinmi@pusan.ac.kr; 4Family Medicine Clinic, Obesity, Metabolism and Nutrition Center, Pusan National University Yangsan Hospital, Yangsan 50612, Republic of Korea; 5Department of Family Medicine and Biomedical Research Institute, Pusan National University Yangsan Hospital, Yangsan 50612, Republic of Korea

**Keywords:** caffeine intake, suicidal ideation, work pattern, sleep duration

## Abstract

Background: Caffeine, a widely consumed stimulant, affects sleep and mental health. Shift work disrupts the circadian rhythm and has been associated with various mental health issues, such as depression, anxiety, and suicidal ideation. Objective: This study explored the associations between caffeine consumption, sleep duration, and mental health outcomes, particularly suicidal ideation, among shift workers in Korea. Methods: Data from the 6th Korea National Health and Nutrition Examination Survey (2013, 2015), which comprised 4723 adults aged 19 and older, were analyzed. Participants were categorized into groups based on average daily coffee consumption and work patterns. Multiple logistic regression analyses were conducted to determine the impact of caffeine consumption and work patterns on mental health outcomes. Results: Participants worked longer hours, reported higher perceived stress levels, and slightly decreased sleep duration as daily coffee consumption increased. However, no significant differences were observed in depression or suicidal ideation across the coffee consumption groups. Logistic regression analysis indicated a trend towards higher suicidal ideation risk with increased coffee intake, particularly among those who consumed three or more cups per day (OR 5.67, 95% CI 1.82–17.59). Conclusion: This study suggests a complex relationship between caffeine consumption, work patterns, and mental health outcomes. Although caffeine intake is associated with increased work hours and stress, its impact on suicidal ideation is influenced by occupational factors.

## 1. Introduction

Caffeine, a widely consumed stimulant, has the ability to enhance alertness, reduce fatigue, and affect the production of several neurotransmitters, such as dopamine and serotonin, which regulate mood and depression [1]. Studies have revealed that caffeine intake close to bedtime can significantly delay sleep onset, reduce total sleep time, and impair sleep efficiency [2]. In addition to its impact on sleep, caffeine consumption has been linked to various mental health outcomes. Moderate caffeine intake is associated with improved mood and cognitive performance; however, excessive or chronic consumption may exacerbate symptoms of anxiety and depression [3].

Shift work has both short- and long-term health consequences owing to its disruption of the circadian rhythm. Disturbances in the sleep–wake cycle and misalignment of the internal biological clock can lead to sleep disorders [4], mental health issues [5], as well as increased risks of breast and other cancers [6]. Observational studies have reported that shift workers experience higher prevalence rates of depression, anxiety, and suicidal ideation compared with day workers [7,8,9].

Shift workers also tend to consume more caffeine compared with non-shift workers to combat daytime sleepiness and fatigue [10]. These findings suggest that circadian disruption may increase the risk of mental health issues and indicate differences in caffeine consumption between shift workers and non-shift workers [11]. However, they do not establish a direct correlation between these factors, emphasizing the need for further investigation.

Therefore, this study investigated the associations between caffeine consumption, sleep duration, and mental health outcomes among shift workers in Korea.

## 2. Methods

### 2.1. Study Population

This study utilized raw data from the first and third years of the 6th Korea National Health and Nutrition Examination Survey (KNHANES, 2013, 2015). Of the 8018 and 7380 participants who completed the surveys in 2013 and 2015, respectively, this study included 9172 adults aged 19 years or older. Of these, 3515 who were being treated for depression, had diseases that could affect depression, or were missing data, were excluded. Furthermore, we excluded 6226 participants aged <19 years, 168 being treated for depression, 1898 who had diseases that affected depression, 281, 301, 839, 401, 40, and 36, with stroke, myocardial infarction and angina, arthritis, cancer diagnosis, renal failure, and liver cirrhosis, respectively. Additionally, 1449 participants did not participate in the food intake frequency survey, and 934 participants were missing data on coffee intake. Therefore, the final sample included 4723 participants. This study was approved by the Institutional Review Board (IRB) of Pusan National University Hospital (IRB No. 2501-009-147 approved on 20 January 2025). Participant consent was waived due to the retrospective nature of the study and the use of anonymized data.

### 2.2. Data Collection

The National Health and Nutrition Survey comprised a physical examination, health survey, and nutrition survey and obtained various information via self-administered questionnaires and interviews with investigators. Participants’ age, education level (elementary school or lower, middle school or lower, middle school or high school, college or higher), income level (income quartile), marital status, smoking status (non-smoker, former, current), alcohol consumption (≤1 a month, 2–4 times a month, 2–3 times a week, ≥4 times a week), comorbidities (hypertension, dyslipidemia, asthma, diabetes, chronic obstructive pulmonary disease, atopic dermatitis, allergic rhinitis), average daily sleep duration, weekly working hours, subjective health status (rated on a 5-point scale, 1 (very unhealthy), 2 (somewhat unhealthy), 3 (average), 4 (somewhat healthy), and 5 (very healthy)) were obtained via a questionnaire. The aerobic physical activity rate consisted of the proportion of participants who had engaged in ≥150 min of moderate-intensity physical activity or ≥75 min of vigorous-intensity physical activity in the previous week. Participants who met these criteria were coded as 1, and those who did not were coded as 0. Perceived stress was assessed by having participants report their usual level of stress using the following options: (1) I feel extremely stressed, (2) I feel quite stressed, (3) I feel slightly stressed, or (4) I feel little to no stress. Participants who reported feeling extremely or quite stressed (responses 1 or 2) were coded as 1, corresponding to a high level of stress perception, while those who reported feeling slightly stressed or experiencing little to no stress (responses 3 or 4) were coded as 0. Depression was assessed by asking participants if they had experienced depressive symptoms for two or more consecutive weeks. Individuals who answered “yes” were coded as 1, while those who answered “no” were coded as 0. Suicidal ideation was assessed by asking “Have you seriously thought about suicide in the past year?” Responses were recorded as “Yes” (coded as 1) or “No” (coded as 0). Suicide plan was assessed by asking “Have you made a specific plan to commit suicide in the past year?” Responses were recorded as “Yes” (coded as 1) or “No” (coded as 0).

Caffeine consumption was assessed based only on coffee intake using a food frequency survey. According to a 2013 survey by the Korea Food and Drug Administration [https://www.mfds.go.kr/brd/m_99/down.do?brd_id=ntc0021&seq=44023&data_tp=A&file_seq=1, accessed on 5 January 2025], 90% of Korean adults’ caffeine consumption came from coffee-related products, such as coffee mixes and prepared coffee. Therefore, caffeine consumption was calculated based on coffee intake. The frequency of coffee consumption was included in the 63-food intake frequency item of the Nutrition Survey, and was classified into nine items: almost never, once/month, 2–3 times/month, once/week, 2–3 times/week, 4–6 times/week, once/day, twice/day, and thrice/day. Coffee consumption was assessed using self-reported data from the Korean National Health and Nutrition Examination Survey and the Food Frequency Survey. Participants reported their average intake over the past year using nine frequency categories: nearly never, 1 time per month, 2–3 times per month, 1 time per week, 2–4 times per week, 5–6 times per week, 1 time per day, 2 times per day, and 3 times per day. The midpoint of each frequency category was used to estimate the daily frequency of consumption. In addition, the serving size per occasion was reported in three categories (1 ts, 2 ts, and 3 ts), where 2 ts represents the standard portion (1 cup). We calculated coffee consumption for a week and divided it by 7 to determine daily coffee intake. Finally, coffee consumption was categorized into four groups: less than 1, less than 2, less than 3, and 3 or more, which correspond to less than 1, less than 2, less than 3, and 3 or more cups per day, respectively.

Respondents reported their work patterns in the following categories: (1) Daytime work (6 AM~6 PM), (2) Evening shift (2 PM–12 AM), (3) Night shift (9 PM–8 AM the next day), (4) Regular day and night shift, (5) 24 h rotating shift, (6) Split shift (working in two or more time slots within a day), (7) Irregular shift, and (8) Other. For analysis, these categories were reclassified into four groups: Daytime (1), Evening–night (2–3), Regular shift (4–5), and Irregular shift (6–8).

### 2.3. Statistical Analysis

The National Health and Nutrition Survey data were sampled via a two-stage stratified probability sampling method, a complex sample design method. Data were analyzed to reflect three elements: weight, strata, and cluster. After participants were classified into four groups based on their daily coffee consumption, their general characteristics (sociodemographic, health-related, and occupation-related factors) were compared. Generalized linear regression analyses and chi-squared (χ^2^) tests were used to compare continuous and categorical variables. Multiple logistic regression was conducted to calculate the odds ratios (ORs) for suicidal ideation according to coffee intake and work patterns. The model was adjusted for the following covariates: sex, age, marital status, education level, income level, smoking status, alcohol consumption frequency, aerobic physical activity rate, comorbidities, sleep duration, and weekly working hours. Statistical analyses were performed via SAS Enterprise Guide 8.2 (SAS Institute Inc., 2019, Cary, NC, USA). All tests were two-sided, with the significance level set as *p* < 0.05.

## 3. Results

To compare the participants’ baseline characteristics according to daily coffee consumption, they were classified into four groups based on average daily coffee intake (Table 1). Of the participants, 55.4% were male, the average age was 39.4 years, and 28.9% were married. As daily coffee consumption increased, the percentage of highly educated people (44.9% → 45.3% → 53.7% → 60.5%; *p* < 0.001) and current smokers (48.2% → 55.0% → 68.3% → 69.7%; *p* < 0.001) increased. However, no significant differences were observed in aerobic physical activity rate or the presence of comorbidities among the four groups.

We analyzed work patterns and mental health status according to average daily coffee consumption (Table 2). Approximately 80% were engaged in daytime work, with an average weekly working time of 42.46 h. As daily coffee consumption increased, the average weekly working hours also increased (40.54 → 53.25 → 43.36 → 44.80 h; *p* < 0.001). Sleep duration also decreased slightly with increasing coffee consumption (6.90 →6.78 → 6.79 → 6.61 h; *p* < 0.001). This suggests that individuals with higher coffee intake tend to work longer hours on average and sleep for shorter durations.

Furthermore, perceived stress levels increased with increased coffee consumption (26.9% → 25.3% → 28.9% → 33.7%; *p* = 0.008); however, no significant differences were observed in the prevalence of two-week depressive episodes or suicidal ideation among the groups.

A logistic regression analysis was performed to analyze the factors that affected suicidal ideation (Table 3). As the average daily coffee intake increased, the risk of suicidal ideation also tended to increase. It increased by 16% in the three or more group compared with the less than 1 group; however, the increase was not statistically significant. In the Model 2 analysis that adjusted for multiple variables, such as sex, age, marital status, education level, income level, current smoker, alcohol consumption frequency, aerobic physical activity rate, and comorbidities, the risk of suicidal ideation significantly increased in the three or more group (OR 5.67, 95% CI 1.82–17.59). This trend was maintained in Model 3 that adjusted for sleep duration (OR 4.67, 95% CI 1.36–16.00). However, the risk was not statistically significant in Models 4 (OR 3.05, 95% CI 0.75–12.46) and 5* (OR 3.08, 95% CI 0.73~13.10), which adjusted for working hours per week and work patterns, respectively.

The risk of suicidal ideation according to work patterns increased by 26% in the evening-night group compared with the daytime group; however, it was not statistically significant. In Models 1–5**, which adjusted for variables in regular and irregular shift groups, the OR of suicidal ideation increased. In the regular shift group, the risk significantly increased in Model 2 (OR 5.77, 95% CI 1.19–27.99), and this trend was maintained in Models 3, 4, and 5**.

## 4. Discussion

This study explored the relationship between daily coffee consumption and mental health outcomes, which included suicidal ideation, in a nationally representative sample of adults in Korea. Our findings provide insights into the complex associations between daily coffee intake, work patterns, and mental health, highlighting the potential roles of demographic and lifestyle factors.

Although a statistical association was observed, causality cannot be confirmed, and the possibility of a reverse relationship should be considered. The results found that as coffee consumption increased, average weekly working hours tended to rise. This suggests that individuals in occupations requiring extended hours and sustained concentration may consume more caffeine. Therefore, rather than caffeine consumption directly affecting mental health, the work environment may influence both caffeine intake and mental health outcomes [10,11].

Previous studies have suggested that caffeine consumption may reduce the risk of depression by enhancing mood through its stimulating effects or promoting wakefulness [12]. However, in this study, higher caffeine intake was associated with an increased risk of suicidal ideation in the adjusted models. After further adjusting for the work environment, this association was no longer statistically significant. This indicates that confounding factors such as working hours and sleep deprivation may hold a prominent place in working conditions and lifestyle factors, rather than caffeine itself, and could be key contributors to mental health outcomes.

We observed an increase in weekly working hours and higher perceived stress levels as daily coffee consumption increased. Increasing work hours per week could lead to increased stressors, which could lead to increased caffeine intake to cope. A positive correlation was observed between work hours per week and caffeine intake. A study found that for every 40 h work week, caffeine intake increased by 15.0 mg (95% 95% CI: 3.42, 26.6; *p* = 0.0012) [13]. This suggests that individuals who work longer hours tend to consume more caffeine. Previous studies also demonstrated a strong association between long working hours and various forms of psychological and physical stress. A study on white-collar workers in Korea found that as working hours increased, the odds of experiencing psychosocial stress responses also increased [14]. Long working hours were significantly associated with higher levels of occupational stress and depression [15]. Meanwhile, increasing perceived stress levels suggested that higher coffee consumption may be associated with greater psychological strain [16], potentially due to the interaction of increased work demands and caffeine’s effects on the central nervous system [17].

Furthermore, we observed an increase in weekly working hours and shorter sleep duration as daily coffee consumption increased. Previous studies revealed that when weekly working hours among shift workers increased, sleep time decreased [16], which negatively impacted suicidal ideation and increased mental illnesses, such as depression and anxiety [18]. Longer work hours often lead to reduced sleep time [19]. Insufficient sleep is strongly associated with increased stress levels [20]. As sleep quality worsens, stress scores tend to increase. This creates a cycle where stress can further disrupt sleep, leading to more stress [21]. Higher caffeine intake is associated with increased anxiety and stress scores [22]. It is important to note that the relationship between caffeine and sleep is not straightforward. Some studies found no significant correlation between average daily caffeine consumption and daily sleeping hours [2,23]. However, the timing and amount of caffeine intake can still impact sleep quality, especially in shift workers [24].

Logistic regression analysis demonstrated a trend toward higher suicidal ideation risk with increased coffee consumption, which became significant, adjusting for confounders such as demographic factors, lifestyle behaviors, and sleep duration. Elevated odds ratio in the three or more group (Model 2: OR 5.67, 95% CI 1.82–17.59) indicated that suicidal ideation may be influenced by other factors, such as education level and smoking, rather than caffeine intake. Current daily smokers had an increased risk of subsequent suicidal thoughts or attempts compared with never smokers [25]. Furthermore, socioeconomic factors, such as lower education levels and financial problems, were associated with a higher risk of suicidal behaviors [26]. Chronic pain and major medical conditions were also associated with higher suicidal ideation [27]; however, these were excluded from this study.

Notably, the statistical significance of this association diminished after adjusting for work patterns and weekly working hours in Models 4 (OR 3.05, 95% CI 0.75–12.46) and 5**(OR 3.08, 95% CI 0.73–13.10), which suggested that occupational factors mediated the relationship between coffee consumption and suicidal ideation [28,29]. Research suggests that occupational stress plays a significant role in increasing the risk of suicidal ideation by elevating depressive symptoms [30].

Analysis by work patterns revealed that the regular shift group exhibited an increased risk of suicidal ideation in the model adjusted for sociodemographic factors compared with the daytime groups. This could be interpreted as the influence of a regular shift group’s sleep duration, weekly working hours, and coffee consumption on the increased risk of suicidal ideation. Further research should examine the reasons for this increased risk.

This study’s strengths include the use of a large, nationally representative sample and the application of robust statistical adjustments for various confounders. However, several limitations should be noted. First, the cross-sectional design precludes causal inferences, highlighting the need for future longitudinal studies to establish temporal relationships. Second, reliance on self-reported data for coffee consumption and mental health outcomes may have introduced reporting bias. Third, caffeine intake was assessed solely by coffee consumption frequency (cups per day) rather than specific caffeine content, preventing precise calculations of caffeine intake. Additionally, caffeine content varies across brands of caffeinated beverages; however, the available dataset lacked information on specific brands or types of coffee consumed, further limiting accuracy. Therefore, we were unable to calculate the exact amount of caffeine consumed. Fourth, caffeine is also present in other foods and beverages, such as cola, green tea, and chocolate, which were not accounted for. This restricts the study’s ability to estimate total caffeine intake comprehensively. Fifth, the results of this study may not be generalizable because missing data were excluded, as were patients with serious comorbidities that could significantly affect mental health. Additionally, while our findings suggest an association between higher coffee consumption and shorter sleep duration, we did not directly assess potential mediating factors such as perceived stress.

Our findings suggest that increased work hours are associated with higher perceived stress, greater caffeine consumption, and slightly shorter sleep duration. Additionally, higher caffeine intake was associated with increased odds of reporting suicidal ideation. However, we did not observe direct associations between sleep and stress, sleep and suicidal ideation, or suicidal ideation and stress, suggesting that these relationships may be independent rather than indicative of a complex interplay. Further research is needed to explore potential underlying mechanisms.

## 5. Conclusions

This study suggests a complex relationship between caffeine consumption, work patterns, and mental health outcomes. While caffeine intake was associated with increased work hours and stress, its impact on suicidal ideation was influenced by occupational factors. Further research, particularly longitudinal studies, is required to better understand the causal mechanisms underlying these associations.

## Figures and Tables

**Table 1 nutrients-17-01155-t001:** Participants’ baseline characteristics according to average daily coffee intake.

			Average Daily Coffee Intake (Cups)				
Variables		Overall * N * = 4723	Less than 1 *N* = 1769	Less than 2 *N* = 1775	Less than 3 *N* = 660	Three or More *N* = 519	* p * Value
Sex							
	Male	55.4 (0.7)	52.6 (1.3)	55.5 (1.2)	63.4 (2.0)	61.1 (2.3)	<0.001
	Female	44.6 (0.7)	47.4 (1.3)	44.5 (1.2)	36.6 (2.0)	38.9 (2.3)	
Age		39.5 (0.2)	37.9 (0.3)	42.8 (0.3)	39.9 (0.5)	38.9 (0.5)	<0.001
Married		28.9 (0.9)	35.7 (1.4)	17.8 (1.2)	24.7 (2.1)	27.5 (2.5)	<0.001
Education level							
	Less than elementary school	4.0 (0.3)	3.7 (0.5)	4.8 (0.6)	4.1 (0.8)	2.8 (0.9)	<0.001
	Middle school	5.8 (0.4)	5.7 (0.7)	7.8 (0.7)	4.5 (0.9)	4.3 (0.9)	
	High school	42.0 (1.0)	45.7 (1.5)	42.1 (1.5)	37.6 (2.4)	32.4 (2.6)	
	College or higher	48.2 (1.1)	44.9 (1.6)	45.3 (1.5)	53.7 (2.5)	60.5 (3.0)	
Income level							
	1st quartile	7.2 (0.5)	9.2 (0.9)	6.0 (0.6)	6.2 (1.0)	3.8 (0.9)	<0.001
	2nd quartile	24.1 (1.0)	26.0 (1.4)	25.5 (1.3)	19.9 (1.8)	19.2 (2.3)	
	3rd quartile	32.4 (1.1)	33.1 (1.6)	32.6 (1.4)	32.4 (2.1)	31.3 (2.6)	
	4th quartile	36.3 (1.3)	31.7 (1.6)	35.9 (1.6)	41.5 (2.5)	45.7 (3.0)	
Current smoker		57.1 (1.2)	48.2 (2.3)	55.0 (2.2)	68.3 (2.7)	69.7 (3.4)	<0.001
Alcohol consumption frequency							
	<=1 time/month	43.4 (0.8)	45.7 (1.4)	41.9 (1.4)	36.9 (2.3)	39.0 (2.6)	0.055
	2–4 times/month	30.6 (0.8)	30.6 (1.4)	31.2 (1.4)	32.4 (2.2)	31.4 (2.6)	
	2–3 times/week	19.7 (0.7)	18.0 (1.1)	19.9 (1.2)	24.0 (2.0)	22.8 (2.2)	
	>=4 times/week	6.3 (0.4)	5.7 (0.7)	7.0 (0.7)	6.7 (1.1)	6.8 (1.3)	
Aerobic physical activity, yes		56.6 (1.2)	57.9 (2.1)	53.5 (2.0)	59.4 (3.1)	60.1 (3.4)	0.220
Comorbidities		30.3 (0.8)	31.3 (1.4)	27.4 (1.2)	31.4 (2.1)	32.6 (2.3)	0.063
	Hypertension	8.1 (0.5)	8.3 (0.8)	9.6 (0.8)	7.1 (1.1)	6.5 (1.3)	0.147
	Dyslipidemia	6.6 (0.4)	6.1 (0.6)	7.3 (0.7)	6.3 (1.2)	8.1 (1.3)	0.421
	Asthma	2.1 (0.3)	2.0 (0.4)	1.2 (0.3)	3.7 (0.9)	3.5 (0.9)	0.002
	Diabetes mellitus	3.2 (0.3)	3.1 (0.5)	3.7 (0.5)	2.4 (0.7)	2.8 (0.8)	0.487
	COPD	0.4 (0.1)	0.1 (0.1)	0.4 (0.2)	0.6 (0.5)	0	NS
	Atopic dermatitis	3.7 (0.3)	3.6 (0.6)	2.2 (0.4)	4.8 (1.0)	5.5 (1.2)	0.008
	Allergic rhinitis	17.7 (0.7)	18.9 (1.2)	14.8 (1.0)	18.7 (1.8)	19.0 (2.1)	0.032

Note. Age is the mean and standard error. The remaining variables are proportions and standard errors. COPD: chronic obstructive pulmonary disease. NS: Not significant

**Table 2 nutrients-17-01155-t002:** Work patterns and mental health status according to average daily coffee consumption.

		Average Daily Coffee Intake (Cups)					
Variables		Overall *N* = 4723	Less than 1 *N* = 1769	Less than 2 *N* = 1769	Less than 3 *N* = 660	Three or More *N* = 519	*p* Value
Work Patterns							
	Daytime	80.2 (0.8)	77.8 (1.4)	81.9 (1.2)	80.3 (2.1)	84.4 (2.0)	0.022
	Evening–night	13.3 (0.6)	15.4 (1.2)	10.9 (1.0)	14.5 (1.8)	9.8 (1.7)	
	Regular shift	4.3 (0.4)	4.1 (0.7)	5.3 (0.7)	4.0 (0.9)	3.3 (1.1)	
	Irregular	2.2 (0.3)	2.7 (0.6)	1.9 (0.4)	1.2 (0.4)	2.5 (0.9)	
Working hours per week		42.46 (0.30)	40.54 (0.53)	43.24 (0.50)	43.36 (0.85)	44.80 (0.91)	<0.001
Sleep duration, hours		6.82 (0.02)	6.90 (0.04)	6.78 (0.03)	6.79 (0.05)	6.61 (0.06)	<0.001
Stress perception		27.5 (0.8)	26.9 (1.3)	25.3 (1.3)	28.9 (2.0)	33.7 (2.2)	0.008
Subjective health status							
	Very unhealthy	5.7 (0.4)	5.6 (0.7)	5.6 (0.7)	5.7 (1.1)	6.4 (1.3)	0.605
	Somewhat unhealthy	32.3 (0.8)	32.8 (1.3)	32.2 (1.3)	29.7 (2.2)	32.1 (2.1)	
	Somewhat healthy	50.6 (0.8)	50.2 (1.4)	51.9 (1.4)	53.3 (2.2)	49.0 (2.4)	
	Somewhat healthy	10.7 (0.5)	10.3 (0.9)	9.7 (0.8)	10.9 (1.3)	12.4 (1.7)	
	Very healthy	0.8 (0.1)	1.1 (0.3)	0.6 (0.2)	0.5 (0.2)	0.2 (0.2)	
Depression, yes		9.6 (0.5)	10.1 (0.8)	7.7 (0.7)	9.2 (1.2)	8.9 (1.6)	0.214
Suicide							
	Ideation, yes	3.2 (0.3)	3.3 (0.5)	3.0 (0.5)	3.3 (0.7)	3.8 (1.0)	0.853
	Plan, yes	0.9 (0.1)	0.6 (0.2)	1.3 (0.3)	0.9 (0.4)	0.8 (0.5)	0.208

**Table 3 nutrients-17-01155-t003:** Logistic regression analysis of suicidal ideation according to coffee consumption and work patterns.

	Unadjusted	Model 1	Model 2	Model 3	Model 4	Model 5*
	OR (95% CI)					
Average daily coffee intake (cups per day)						
Less than 1	1 (Reference)					
Less than 2	0.90 (0.59–1.39)	0.89 (0.58–1.38)	1.65 (0.49–5.56)	1.50 (0.43–5.21)	1.49 (0.39–5.68)	1.60 (0.40–6.49)
Less than 3	1.02 (0.62–1.70)	1.08 (0.64–1.81)	1.54 (0.46–5.12)	1.41 (0.42–4.76)	1.31 (0.37–4.69)	1.27 (0.33–4.95)
Three or more	1.16 (0.64–2.12)	1.22 (0.67–2.23)	5.67 (1.82–17.59)	4.67 (1.36–16.00)	3.05 (0.75–12.46)	3.08 (0.73–13.10)
	Unadjusted	Model 1	Model 2	Model 3	Model 4	Model 5**
Work patterns						
Daytime	1 (Reference)					
Evening-night	1.26 (0.76, 2.10)	1.27 (0.75, 2.14)	0.60 (0.21, 1.74)	0.59 (0.20, 1.75)	0.57 (0.21, 1.49)	0.60 (0.22, 1.61)
Regular shift	0.89 (0.27, 2.88)	0.96 (0.29, 3.10)	5.77 (1.19, 27.99)	6.57 (1.38, 31.42)	5.95 (1.42, 24.89)	6.35 (1.54, 26.22)
Irregular	0.51 (0.12, 2.16)	0.52 (0.12, 2.21)	2.43 (0.27, 21.92)	1.92 (0.20, 19.00)	1.18 (0.10, 14.70)	1.20 (0.08, 19.10)

Note. OR; odds ratio. Model 1, adjusted for sex and age. Model 2, adjusted for Model 1 + married, education level, income level, current smoker, alcohol consumption frequency, aerobic physical activity, and comorbidities. Model 3, adjusted for Model 2 + sleep duration. Model 4, adjusted for Model 3 + working hours per week. Model 5* adjusted for Model 4 + work patterns. Model 5** adjusted for Model 4 + average daily coffee intake.

## Data Availability

Data and materials are available upon reasonable request. The raw KNHANES data used in this paper can be accessed via the following website: https://www.data.go.kr/data/15076556/fileData.do, accessed on 20 November 2024.

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
