# Peer review of "Mental Health, Sleep, and Caffeine Intake Among Shift Workers in a Nationally Representative Sample of the Korean Adult Population"

_nutrients, 2025, doi:10.3390/nu17071155_

Round 1

Reviewer 1 Report

Comments and Suggestions for Authors

The manuscript entitled "Mental Health, Sleep, and Caffeine Intake among Shift Workers in a Nationally Representative Sample of the Korean Adult Population" reports a study on the associations between caffeine consumption, sleep duration, and mental health outcomes, particularly suicidal ideation, among shift workers in Korea. 

The authors reported that regression analysis the coffee consumption groups. Logistic regression analysis indicated a trend towards higher suicidal ideation risk with increased coffee intake, particularly among those who consumed >4 cups per day. Even more, the authors concluded there is a complex relationship between caffeine consumption, work patterns, and mental health outcomes caffeine intake is associated with increased work hours and stress, occupational factors influence its impact on suicidal ideation.

Even though it is a very complex study and relating caffeine to suicidal ideation is just a possibility and perhaps it is linked much more to the associated factors of habits that are interconnected and have no direct cause but only by a connection of habits, the study presents results that can be used for new promising studies.

Please,  I strongly suggest the authors discuss that even though there is a statistical relationship, it is likely that caffeine has no ties with mental problems or even suicidal ideation. The relationship may even be inverse, as is the case with people involved in jobs that require many hours of concentration who drink a lot of coffee (caffeine). A more in-depth discussion of this point will give better visibility to the results. This point also needs to be added to the abstract and conclusion. 

Please, the authors need to discuss that even though there is a statistical relationship, it is likely that caffeine has no relationship with mental problems or even suicidal ideation. The relationship may even be inverse, as is the case with people involved in jobs that require many hours of concentration who drink a lot of coffee (caffeine). A more in-depth discussion of this point will give better visibility to the results. This point also needs to be added to the abstract and conclusion. 

Author Response

Responses to Reviewer Comments

We thank the reviewers for their thoughtful suggestions and insights, which have enriched the manuscript and helped produce a better and more balanced account of the research. All of our revisions have been highlighted in yellow in the main manuscript.

Reviewer 1: Comment

Comment 1: Please, the authors need to discuss that even though there is a statistical relationship, it is likely that caffeine has no relationship with mental problems or even suicidal ideation. The relationship may even be inverse, as is the case with people involved in jobs that require many hours of concentration who drink a lot of coffee (caffeine). A more in-depth discussion of this point will give better visibility to the results. This point also needs to be added to the abstract and conclusion. 

Response 1: We appreciate the reviewer’s insightful comment. In response, we have revised the Discussion section to explicitly acknowledge that the observed statistical association does not imply a causal relationship between caffeine intake and mental health outcomes. Additionally, we now discuss the possibility of a reverse relationship, where individuals with higher cognitive or occupational demands may consume more caffeine, rather than caffeine directly affecting mental health. Specifically, we have incorporated the following changes to the Results and Discussion sections:

Results section, page 4

As daily coffee consumption increased, the average weekly working hours also increased (40.54 53.25 43.36 44.80 hours; p < 0.001). Sleep duration also revealed a similar trend, and increased with coffee consumption (10.2 10.6 12.1 17.0 hours; p < 0.001). This suggests that individuals with higher coffee intake tend to work longer hours on average and sleep for longer durations.

Discussion section, page 7

Although a statistical association was observed, causality cannot be confirmed, and the possibility of a reverse relationship should be considered. The results found that as coffee consumption increased, average weekly working hours tended to rise. This suggests that individuals in occupations requiring extended hours and sustained concentration may consume more caffeine. Therefore, rather than caffeine consumption directly affecting mental health, the work environment may influence both caffeine intake and mental health outcomes [10, 11].

Previous studies have suggested that caffeine consumption may reduce the risk of depression by enhancing mood through its stimulating effects or promoting wakefulness [26]. However, in this study, higher caffeine intake was associated with an increased risk of suicidal ideation in the adjusted models. After further adjusting for work environment, this association was no longer statistically significant. This indicates that confounding factors such as work intensity, stress, and sleep deprivation may play a more critical role, suggesting that working conditions and lifestyle factors, rather than caffeine itself, could be key contributors to mental health outcomes.

Additionally, we have included this information in the Abstract and Conclusion sections to enhance the clarity and applicability of our findings. The revised Abstract now acknowledges that the relationship between caffeine intake and mental health may be influenced by occupational and lifestyle factors:

Although caffeine intake is associated with increased work hours and stress, its impact on suicidal ideation is influenced by occupational factors.

Similarly, the Conclusion emphasizes the importance of cautious interpretation, considering potential confounders such as work stress and sleep deprivation.

Reviewer 2 Report

Comments and Suggestions for Authors

As we know, caffeine affects sleep and mental health. The paper (nutrients-3461922-peer-review-v1) submitted by Gyu Lee Kim et al.: Mental Health, Sleep, and Caffeine Intake among Shift Workers in a Nationally Representative Sample of The Korean Adult Population. This study focuses on the associations between caffeine consumption, sleep duration, and mental health outcomes, particularly suicidal ideation, among shift workers in Korea. Overall, the paper’s overall structure is highly coherent. However, some issues are suggested to be addressed before publication. See my comments below to improve this manuscript.

1.      The intake of caffeine of everyone is vague. Different brands of caffeine drinks contain different caffeine.

2.      Other foods may also contain caffeine, how to count the caffeine intake from these foods?

3.      What is the standard of stress perception? How to measure stress perception?

4.      How were the other dietary factors excluded?

5.      How to remove the exception value? Please specify this description in the article.

In light of these issues, the manuscript does not meet the journal's standards for publication. We encourage the authors to address these concerns.

Author Response

Responses to Reviewer Comments

We thank the reviewers for their thoughtful suggestions and insights, which have enriched the manuscript and helped produce a better and more balanced account of the research. All of our revisions have been highlighted in yellow in the main manuscript.

Reviewer 2: Comments

Comment 1: The intake of caffeine of everyone is vague. Different brands of caffeine drinks contain different caffeine.

Response 1: We agree with this comment. Caffeine content varies across brands of caffeinated beverages, and this study did not reflect the exact caffeine intake. We have addressed this by revising the manuscript as follows:

Discussion section, page 8

Third, caffeine intake was assessed solely on coffee consumption frequency (cups per day) rather than specific caffeine content, preventing precise calculations of caffeine intake. Additionally, caffeine content varies across brands of caffeinated beverages; however, the available dataset lacked information on specific brands or types of coffee consumed, further limiting accuracy. Therefore, we were unable to calculate the exact amount of caffeine consumed.  

Comment 2: Other foods may also contain caffeine, how to count the caffeine intake from these foods?

Response 2: Thank you raising this pertinent concern. As mentioned, other foods such as cola, green tea, and chocolate may also contain caffeine, and this study did not include these other foods in the analysis.

According to a press release by the South Korean Ministry of Food and Drug Safety on August 6, 2013 (https://www.mfds.go.kr/brd/m_99/down.do?brd_id=ntc0021&seq=44023&data_tp=A&file_seq=1), the distribution of caffeine intake among Korean adults was 71% from coffee mix, 17% from coffee extract, 4% from coffee beverages, 4% from carbonated beverages, and 4% from other sources. Following this, we used coffee as the primary source of caffeine in Korean adults and measured their intake in cups per day for this study. Accordingly, we have revised the manuscript and described the additional research as part of the limitations. Changes to the manuscript are described below:

Methods: Data Collection section, page 3

Caffeine consumption was assessed based on coffee intake using a food frequency survey. According to a 2013 Korea Food and Drug Administration survey [https://www.mfds.go.kr/brd/m_99/down.do?brd_id=ntc0021&seq=44023&data_tp=A&file_seq=1], 90% of Korean adults' caffeine consumption came from coffee-related products, such as coffee mix and prepared coffee. Therefore, caffeine consumption was calculated based on coffee intake.

Discussion section, page 8

Fourth, caffeine is also present in other foods and beverages, such as cola, green tea, and chocolate, which were not accounted for. This restricts the study’s ability to estimate total caffeine intake comprehensively. Finally, the results of this study may not be generalizable because missing data were excluded, as were patients with serious comorbidities that could significantly affect mental health. These limitations should be considered when interpreting the findings.

Comment 3: What is the standard of stress perception? How to measure stress perception?

Response 3: We appreciate the reviewer's insightful comment. Standard measures of stress perception include the Perceived Stress Scale (PSS) and the Perceived Stress Questionnaire (PSQ). However, the Korea National Health and Nutrition Examination Survey did not use these questionnaires. Instead, stress perception was assessed through a self-reported survey, which measured the percentage of individuals aged 19 or older who reported experiencing considerable stress in daily life. For our analysis, we used data based on the percentage of respondents who selected “I feel a lot” in response to stress-related questions. We have revised the manuscript accordingly.

Methods: Data Collection section, page 2

…perceived stress (percentage who “feel a lot” of stress and “feel a great deal” of stress in their daily lives)…

Comment 4: How were the other dietary factors excluded?

Response 4: We understand your comment regarding the exclusion of other dietary factors. You are correct that diet can affect mental health. To address this we first looked at dietary factors that can affect stress and depression such as tryptophan, vitamin B6, magnesium, dietary fiber, carbohydrates, protein, and sodium intake. Among these, we included and analyzed factors such as dietary fiber, carbohydrates, protein, and sodium intake, as these could be analyzed within the scope of the study. We found no significant change in the results for the basic characteristics and regression analysis. The analysis results are given below (Tables 1 & 2).

Table 1. Basic characteristics of dietary factors based on caffeine consumption

Average daily coffee intake (cups)

Overall

N=5,657

<1

<2

<3

>4

P value

Daily consumption, Mean (SE)

Carbohydrate (g)

326.09 (2.40)

311.23 (3.90)

333.93 (3.74)

340.30 (5.75)

341.33 (7.95)

<0.001

Protein (g)

80.32 (0.93)

78.64 (1.45)

80.54 (1.81)

84.17 (2.05)

83.01 (2.46)

0.100

Fat (g)

54.38 (0.71)

53.07 (1.17)

53.30 (1.17)

55.54 (1.69)

58.87 (1.95)

0.060

Energy (Kcal)

2266.05 (18.22)

2191.99 (29.92)

2295.26 (29.54)

2349.13 (42.86)

2368.19 (54.44)

0.001

Sodium (mg)

4448.56 (59.90)

4195.92 (79.69)

4644.01 (133.42)

4767.26 (123.92)

4540.69 (132.33)

<0.001

Dietary fiber (g)

24.24 (0.24)

23.66 (0.38)

25.10 (0.37)

24.59 (0.51)

23.46 (0.59)

0.020

Table 2. Logistic regression analysis of suicidal ideation according to coffee consumption and work pattern.

Model6

OR (95% CI); p-value

Average daily coffee intake

<1

1 (Reference)

<2

1.56 (0.38, 6.37); 0.538

<3

1.10 (0.25, 4.75); 0.902

<4

2.60 (0.57, 11.77); 0.214

Work pattern

Daytime

1 (Reference)

Evening-night

0.52 (0.18, 1.56); 0.244

Regular shift

5.95 (1.28, 27.69); 0.023

Irregular

0.65 (0.02, 16.91); 0.796

Note: Model6-adjusted for Model5 + daily intake of carbohydrate, protein, fat, energy, sodium, and dietary fiber

Comment 5: How to remove the exception value? Please specify this description in the article. “They are asking for clarification on how to determine whether the collected data meets the required criteria during the data collection process. Specifically, the reviewer is requesting details on the criteria used to assess the data. Please include this information in the manuscript.”

Response 5: We acknowledge the reviewer's concern that caffeine content was not measured accurately in this study. Caffeine intake was estimated based on coffee consumption frequency (in cups per day), without accounting for other dietary sources of caffeine. This issue has been addressed in both the Methods and Discussion sections of the manuscript. Please refer to our responses to the Comments 1, 2, and 3 for a detailed explanation.

Round 2

Reviewer 2 Report

Comments and Suggestions for Authors

The authors have carefully revised the content of the article and it basically meets the criteria for acceptability. Therefore, I would like to recommend it for publication in this journal.

Author Response

Reply to Academic Editor

We thank the reviewers for their thoughtful suggestions and insights, which have enriched the manuscript and helped produce a better and more balanced account of the research. All of our revisions have been highlighted in yellow in the main manuscript.

Introduction

  1. It is stated (lines 56-58) that shift work, caffeine and mental health are inter-related but do not cite an appropriate study examining this. The previous statements suggest that circadian disruption might lead to increased risk of mental health issues and that there are differences in caffeine consumed between shift workers and non-shift workers.  This does not suggest a correlation between these factors.

→ Thank you for your insightful feedback. Based on your comments, we have revised the sentence to avoid implying a direct correlation between these factors. The updated version is as follows:

Revised:"These findings suggest that circadian disruption may increase the risk of mental health issues and indicate differences in caffeine consumption between shift workers and non-shift workers. However, they do not establish a direct correlation between these factors, enphasizing the need for further investigation."

This revision ensures that our statement remains cautious and does not overstate the relationships between these variables. Please let us know if you have any further suggestions.

Methods

  1. Please update the statements regarding perceived stress (lines 89-90).  It is not clear how those who experienced lower levels of stress were included in the study.  Was perceived stress measured on a scale?  If so, what was this scale?  Why were only those with high levels of stress included?  I would expect this variable to be categorised similarly to others (e.g., health status).

→ Thank you for your insightful feedback. There are questionnaires such as the Perceived Stress Scale (PSS) and the Perceived Stress Questionnaire (PSQ) as standards of stress perception. In our study, we used the perceived stress variable (mh_stress) from the Korean National Health and Nutrition Examination Survey (KNHANES). Perceived stress is defined based on self-reported responses, where participants indicated their usual level of stress as follows: 1) I feel extremely stressed, 2) I feel quite stressed, 3) I feel slightly stressed, or 4) I feel little to no stress. It is coded as 1 for individuals who reported feeling extremely or quite stressed (responses 1 or 2) and 0 for those who reported feeling slightly stressed or little to no stress (responses 3 or 4).

Revised: "Perceived stress was assessed  by having participants report their usual level of stress as follows: 1) I feel extremely stressed, 2) I feel quite stressed, 3) I feel slightly stressed, or 4) I feel little to no stress. The responses were coded as follows: 1 for individuals who reported feeling extremely or quite stressed (responses 1 or 2) and 0 for those who reported feeling slightly stressed or little to no stress (responses 3 or 4)."

  1. The word ‘percentage’ does not seem correct for the physical activity variable captured in the survey (line 85).  Please update the category descriptions to be much clearer.  

→ Thank you for your insightful feedback. In our study, we used the aerobic physical activity rate variable (pa_aerobic) from the Korean National Health and Nutrition Examination Survey (KNHANES). This variable reflects the proportion of individuals engaging in at least 150 minutes of moderate-intensity or 75 minutes of vigorous-intensity activity in the past week. Individuals meeting these criteria are coded as 1, and those who do not are coded as 0.

Revised: "aerobic physical activity rate reflects the proportion of individuals who engaged in at least 150 minutes of moderate-intensity or 75 minutes of vigorous-intensity physical activity in the past week. Participants who met these criteria were coded as 1, while those who did not were coded as 0."

This revision ensures that the description accurately reflects the definition used in the KNHANES dataset.

Please let us know if you have any further suggestions.

  1. The word ‘percentage’ does not seem correct for the depression variable captured in the survey.  Presumably, participants could respond ‘yes’ or ‘no’.  (lines 91-94).  The current wording suggests only those who responded yes were included in the present study.

→ Thank you for your insightful feedback. Based on your comments, we have revised the sentence. Depression is defined as the percentage of individuals who answered 'yes' (coded as 1) to experiencing depressive symptoms for two or more consecutive weeks. The updated version is as follows:

Revised: depression (percentage of people who answered “yes” or “no”)

Please let us know if you have any further suggestions.

  1. What ‘questionnaire’ was used to capture suicidal ideation and suicide planning?  Was this questionnaire validated?  Can brief details be provided?  How were these data captured (categories?)?

→ Thank you for your insightful feedback. Suicidal ideation and suicide plan were assessed using a simple self-reported questionnaire item. Participants were asked the following question:

Suicidal ideation was asked, "Have you seriously thought about suicide in the past year?"

Responses were recorded as "Yes" or "No."

Suicide plan was asked, “Have you made a specific plan to commit suicide in the past year?” Responses were recorded as “Yes” or “No.” In Table 1, we present the proportion of participants who responded "Yes."

This questionnaire item is commonly used in national health surveys, including the Korean National Health and Nutrition Examination Survey (KNHANES), and has been widely utilized in previous studies on suicidal ideation. However, we acknowledge that it is a single-item measure and does not constitute a comprehensive validated scale for suicidal ideation assessment.

If further details are needed, we would be happy to provide additional clarification.

Revised: "suicidal ideation ("Have you seriously thought about suicide in the past year?" Responses were recorded as "Yes" or "No."), and suicide plan (“Have you made a specific plan to commit suicide in the past year?” Responses were recorded as “Yes” or “No.”)"

  1. I recommend stating that the study captured caffeine consumption from coffee intake only. (line 100)

→ Thank you for your insightful feedback. Based on your comments, we have revised the sentence. The updated version is as follows:

Revised: "Caffeine consumption was assessed based on only coffee intake using a food frequency survey."

  1. The caffeine consumption calculations are not clear (lines 100-104).  Were there two questions in the survey: frequency and number of cups?  If so, please state this more clearly.

→ Thank you for your comment. To clarify, the survey included two separate questions related to coffee consumption.

We hope this clarifies your question. Please let us know if any further details are needed.

Revised: The detailed calculations are as follows: Coffee intake frequency (FF_COFFEE) was recorded on a scale from 1 (rarely consume) to 9 (three times a day), while the typical amount consumed per serving (FA_COFFEE) was categorized as 1 (half a cup), 2 (one cup), and 3 (two cups). To standardize intake, FF_COFFEE was converted into weekly consumption frequency (FQ_COFFEE), ranging from 0 to 27 times per week. The estimated daily coffee intake (TOT_COFFEE) was then calculated as: [TOT_COFFEE = (FQ_COFFEE / 7) × FA_COFFEE] Finally, TOT_COFFEE was categorized into four groups: [0, 1), [1, 2), [2, 3), and [3, 30), which correspond to less than 1, less than 2, less than 3, and three or more cups per day, respectively.

  1. Please change “Work type was investigated…” (line 105) to “Respondents reported their work type in the following categories: daytime work…”  Presumably, these initial categories were based on question(s) in the survey and subsequently re-categorised into four groups.

→ Thank you for your valuable feedback. Based on your suggestion, we have revised the sentence as follows:

Revised: "Respondents reported their work type in the following categories: (1) Daytime work (6AM~6PM), (2) Evening shift (2 PM–12 AM), (3) Night shift (9 PM–8 AM the next day), (4) Regular day and night shift, (5) 24-hour rotating shift, (6) Split shift (working in two or more time slots within a day), (7) Irregular shift, and (8) Other. For analysis, these categories were reclassified into four groups: Daytime (1), Evening-night (2–3), Regular shift (4–5), Irregular shift (6–8)."

This revision ensures clarity and aligns with the classification used in the study.

Please let us know if further modifications are needed.

  1. It is crucial that shift work is appropriately defined in a study exploring this concept.  The categorisation for the work types is not clear.  

→ Thank you for your valuable feedback. We reclassified shift work based on work types (variable working hours) collected from the Korean National Health and Nutrition Survey as replied to comment 8.

Please let us know if further modifications are needed.

  1. How were the hours of work determined?  The first part of this paragraph suggests categorical survey responses and the second suggests that work schedules were available.  Please provide sufficient detail.

→ In this study, we were not to utilize the occupational classification according to the standard occupational classification in detail.

  1. What was considered ‘daytime’?  Was this 9am-5pm or some other schedule?

→ Thank you for your valuable feedback. In response to the question, “Do you work mainly during the day (between 6 AM and 6 PM) or at other times?”, the respondents answered 1. Mostly during the day, 2. Evening shift (between 2 PM and midnight), 3. Night shift (between 9 PM and 8 AM the next day), 4. Regular day and night shifts, 5. 24-hour shifts, 6. Split shifts (more than two work periods per day), 7. Irregular shifts, 8. Other.

We have revised the manuscript to reflect the comments in response to comment 8.

Please let us know if further modifications are needed.

  1. How many hours of work are needed to occur between 2pm-12am for an evening shift to be categorised as such?  The same needs to be considered for night work and rotating shift work.  

→ Thank you for your valuable feedback. In our study, we did not categorize work types based on the number of hours worked per shift. Instead, work type classification was based on self-reported responses regarding the primary work schedule.

Regarding weekly working hours, the 2013 and 2015 Korean National Health and Nutrition Examination Survey (KNHANES) collected data using the following question:

"How many hours per week, including overtime but excluding meal breaks, do you typically work at your job?"

Participants responded with the total number of hours worked per week.

If necessary, we can clarify this further. Please let us know if you have any additional concerns.

  1. What was classed as rotating shift work?  Was the rotating shift work category based on the survey responses?

→ Thank you for your question. The rotating shift work category in our study was based on self-reported survey responses regarding work type. In the Korean National Health and Nutrition Examination Survey (KNHANES), participants selected their work type from the following categories:

1.Daytime work

2.Evening shift (2 PM–12 AM)

3.Night shift (9 PM–8 AM the next day)

4.Regular day and night shift

5.24-hour rotating shift

6.Split shift (working in two or more time slots within a day)

7.Irregular shift

8.Other

For analysis, we reclassified work types into four groups:

Daytime: (1)

Evening-night: (2–3)

Regular shift (Rotating shift work): (4–5)

Irregular shift: (6–8)

In this classification, rotating shift work included workers with regular day and night shifts and 24-hour rotating shifts (categories 4 and 5). This classification was directly based on the survey responses.

We hope this clarifies your question. Please let us know if any further details are needed.

  1. There seems to be a misunderstanding about what constitutes shift work.  Shift is not only rotating schedules.

→ Thank you for your clarification. We acknowledge that shift work is a broad term that includes not only rotating schedules but also evening shifts, night shifts, and other non-standard work schedules.

In our study, shift work was classified based on self-reported survey responses. Participants selected their work type from predefined categories, including evening shifts, night shifts, and various types of rotating shifts. For analysis, we grouped these into broader categories, with rotating shift work specifically referring to regular day and night shifts and 24-hour rotating shifts.

If needed, we can further clarify our classification in the manuscript. Please let us know if you have any specific suggestions.

  1. What categories comprised regular day work?  What categories comprised night rotating (as opposed to 3-shift rotating)?  What categories comprised split shifts?  What categories comprised irregular rotating?  This is not clear from your descriptions.

→ Thank you for your feedback. We appreciate the opportunity to clarify our classification of work types. The categories were defined based on self-reported responses from the Korean National Health and Nutrition Examination Survey (KNHANES), and for analysis, we reclassified them into broader groups as follows:

  • Regular day work: Included participants who reported working primarily during the daytime.
  • Night rotating shift (as opposed to 3-shift rotating): Included those who reported regular day and night rotating shifts (i.e., predictable shifts alternating between daytime and nighttime).
  • Split shifts: Included those who reported split shifts, defined as working in two or more separate time slots within a single day.
  • Irregular rotating shifts: Included participants who reported irregular shift patterns, such as those with unpredictable or varying work schedules.

We recognize that our initial description may not have been sufficiently detailed, and we appreciate your request for clarification. If needed, we are happy to further elaborate in the manuscript.

  1. Why was perceived stress not included in the regression models?

→ Thank you for your insightful comment. We conducted regression analyses including perceived stress as an response variable; however, we did not include these results in the manuscript.

For your reference, we have attached the results of the regression models incorporating perceived stress. If you believe this information should be included in the manuscript, we would be happy to make the necessary revisions.

Please let us know if you have any further suggestions.

  1. What was the rationale for running the different models (models 1-5)?  What was the rationale for including the specific covariates?  These seem somewhat arbitrarily selected for each model.

→ Thank you for your valuable feedback. The rationale for constructing Models 1-5 was to systematically examine the associations while accounting for potential confounders in a stepwise manner.

  • Model 1: Included only the primary independent variable to assess the crude association.
  • Model 2: Adjusted for basic demographic variables (e.g., age, sex) to control for fundamental confounding effects.
  • Model 3: Further adjusted for socioeconomic factors (e.g., education level, income) as these are known to influence both the exposure and outcome.
  • Model 4: Included health-related behaviors (e.g., smoking, alcohol consumption, physical activity) to account for lifestyle factors that could mediate or confound the association.
  • Model 5: Adjusted for additional clinical variables (e.g., comorbidities) to provide a comprehensive analysis while minimizing residual confounding.

The selection of covariates was based on previous literature and theoretical considerations, ensuring that key confounders were included in a structured manner. However, we acknowledge that alternative modeling approaches could be considered, and we appreciate any further recommendations you may have.

Results

  1. Please ensure Table 1 has sensible categories for coffee consumption.  There is currently no category that captures ‘4’.  Presumably, <1 = 0 cups.  If so, I would suggest changing the category labels.

→ Thank you for your feedback. To ensure clearer categorization of coffee consumption, we have revised the categories in Table 1 as follows:

  • Less than 1 cup per day
  • Less than 2 cups per day
  • Less than 3 cups per day
  • Three or more cups per day

This revision provides a more logical grouping of coffee consumption levels, ensuring that all values are properly categorized. We appreciate your suggestion and hope this adjustment improves clarity. Please let us know if you have any further recommendations.

  1. An increase in work hours with increasing cups of coffee does not only suggest that those who consume more coffee tend to work longer hours.  Instead, those who work longer hours may consume more coffee.  The same applies for sleep duration.  The statement on lines 148-149 would be better placed in the discussion.

→ Thank you for your valuable feedback. While reviewing the statistical results, we found an error in the reported sleep duration values. The previous version of Table 2 mistakenly presented the proportion of participants sleeping less than six hours instead of the actual mean sleep duration.

We have now corrected the table as follows:Previous (incorrect values for <6 hours sleep category):10.2 (0.8) → 10.6 (0.9) → 12.1 (1.4) → 17.0 (1.9), p < 0.001

Revised (correct mean sleep duration values) Table 2 :

  6.90 (0.04) → 6.78 (0.03) → 6.79 (0.05) → 6.61 (0.06), p < 0.001

Accordingly, we have revised the results and discussion sections to accurately reflect this trend. The corrected statement now reads:

Revised (correct mean sleep duration values) Results: "Sleep duration also revealed a similar trend, decreasing slightly with increasing coffee consumption (6.90 → 6.78 → 6.79 → 6.61 hours; p < 0.001). This suggests that individuals with higher coffee intake tend to work longer hours on average and sleep for shorter durations."

We appreciate your careful review and the opportunity to correct this mistake. Please let us know if you have any further comments.

Discussion

There are a number of aspects of the discussion that warrant a more careful and thorough examination.  Please consider how the discussion may more accurately reflect the study finding and discuss the relevant issues without overstating what was found.  The section also needs some editing for clarity.

  1. Bi-directionality needs to be mentioned earlier in the section and discussed more thoroughly.

→ Thank you for your valuable feedback. We acknowledge the importance of addressing the potential bidirectional relationships in our discussion. To improve clarity, we have moved this discussion to an earlier part of the section and expanded it to provide a more thorough examination of possible reverse causality.

We appreciate your suggestion and believe this revision strengthens the overall interpretation of our findings. Please let us know if you have any further recommendations.

  1. It is unusual that higher caffeine consumption is associated with longer sleep.  Please discuss in more detail.  Is there evidence of heightened stress or is this speculation?

→ Thank you for your insightful comment. Upon re-examining our results, we found an error in the reported sleep duration values. The previous version of Table 2 mistakenly presented the proportion of participants sleeping less than six hours instead of the actual mean sleep duration.

We have now corrected the table as follows: Previous (incorrect values for <6 hours sleep category):10.2 (0.8) → 10.6 (0.9) → 12.1 (1.4) → 17.0 (1.9), p < 0.001

Revised (correct mean sleep duration values):6.90 (0.04) → 6.78 (0.03) → 6.79 (0.05) → 6.61 (0.06), p < 0.001

Accordingly, we have revised the results and discussion sections to reflect that higher coffee consumption is actually associated with slightly shorter sleep duration, which aligns more closely with existing literature.

Regarding the potential role of stress, while we did not include perceived stress in the primary analysis, previous studies suggest that individuals under chronic stress may experience disrupted sleep patterns and increased caffeine intake as a coping mechanism. However, without direct measures of stress in our analysis, we cannot conclusively determine whether stress mediates this relationship. We acknowledge this as a limitation and will briefly address it in the discussion. We appreciate your careful review and the opportunity to correct and clarify this aspect of our study. Please let us know if you have any further suggestions.

Revised: "Additionally, while our findings suggest an association between higher coffee consumption and shorter sleep duration, we did not directly assess potential mediating factors such as perceived stress."

  1. In my mind, the results do not suggest that caffeine, suicidal ideation and work type are inter-related and your comments about this seem a stretch.  You have observed a number of associations, each of which may be independent from each other.  Your results suggest that increased work hours were associated with (a) higher perceived stress, (b) higher coffee consumption, (c) longer sleep duration.  Further, increased coffee consumption was associated with an increased odds of reporting suicidal ideation.  There are no observed associations between sleep and stress, sleep and suicidal ideation or suicidal ideation and stress and therefore, no strong evidence of complex interrelationships.  I recommend tempering your statements accordingly.

→ Thank you for your insightful feedback. We acknowledge your concern regarding the interpretation of the relationships between caffeine consumption, suicidal ideation, and work type. Upon review, we have revised our discussion to ensure that our statements more accurately reflect the observed associations without overinterpreting their interconnections.

Our results indicate that:

  • Increased work hours were associated with (a) higher perceived stress, (b) higher coffee consumption, and (c) slightly shorter sleep duration (corrected from the previous version).
  • Higher coffee consumption was associated with an increased odds of reporting suicidal ideation.
  • However, no significant associations were found between sleep and stress, sleep and suicidal ideation, or suicidal ideation and stress.

Based on these findings, we have tempered our discussion to reflect that while we observed multiple associations, each of these may be independent rather than part of a complex interrelationship. We appreciate your suggestion and have adjusted our conclusions accordingly to avoid overstatement. Please let us know if you have any further recommendations.

Revised:"Longer work hours often lead to reduced sleep time [19]. Insufficient sleep is strongly associated with increased stress levels [20]. As sleep quality worsens, stress scores tend to increase. This creates a cycle where stress can further disrupt sleep, leading to more stress [21]. Higher caffeine intake is associated with increased anxiety and stress scores [22]. It's important to note that the relationship between caffeine and sleep is not straightforward. Some studies found no significant correlation between average daily caffeine consumption and daily sleeping hours [23,24]. However, the timing and amount of caffeine intake can still impact sleep quality, especially in shift workers [25]."

  1. Take care to use consistent terminology with respect to your work type groups.

→ Thank you for your helpful feedback. We have carefully reviewed the manuscript and ensured that consistent terminology is used when referring to work type groups. Specifically, we have standardized the term to "work patterns" throughout the text, tables, and figures for clarity and consistency. We appreciate your attention to detail and believe this revision improves the overall readability of our findings. Please let us know if you have any further suggestions.
